# Learning to Walk in Minutes Using Massively Parallel Deep Reinforcement Learning

**Nikita Rudin**
ETH Zurich and NVIDIA
rudinn@ethz.ch

**David Hoeller**
ETH Zurich and NVIDIA
dhoeller@ethz.ch

**Philipp Reist**
NVIDIA
preist@nvidia.com

**Marco Hutter**
ETH Zurich
mahutter@ethz.com

**Abstract:** In this work, we present and study a training set-up that achieves fast policy generation for real-world robotic tasks by using massive parallelism on a single workstation GPU. We analyze and discuss the impact of different training algorithm components in the massively parallel regime on the final policy performance and training times. In addition, we present a novel game-inspired curriculum that is well suited for training with thousands of simulated robots in parallel. We evaluate the approach by training the quadrupedal robot ANYmal to walk on challenging terrain. The parallel approach allows training policies for flat terrain in under four minutes, and in twenty minutes for uneven terrain. This represents a speedup of multiple orders of magnitude compared to previous work. Finally, we transfer the policies to the real robot to validate the approach. We open-source our training code to help accelerate further research in the field of learned legged locomotion: https://leggedrobotics.github.io/legged_gym/.

**Keywords:** Reinforcement Learning, Legged Robots, Sim-to-real

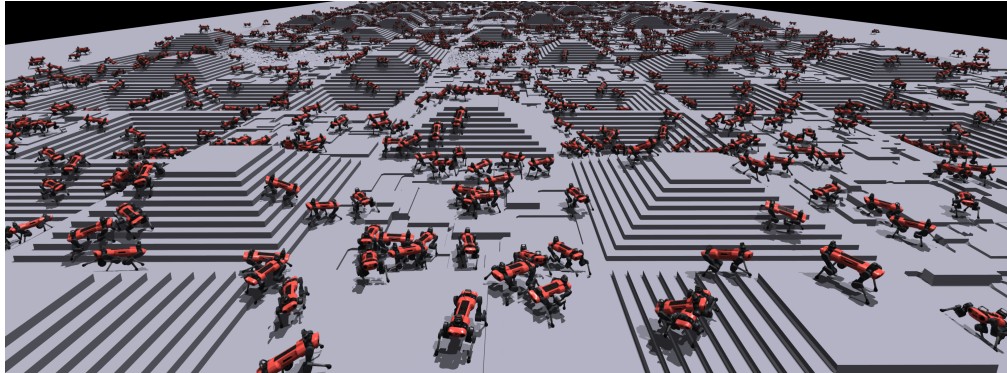

Figure 1: Thousands of robots learning to walk in simulation.

## 1 Introduction

Deep reinforcement learning (DRL) is proving to be a powerful tool for robotics. Tasks such as legged locomotion [1], manipulation [2], and navigation [3], have been solved using these new tools, and research continues to keep adding more and more challenging tasks to the list. The amount of data required to train a policy increases with the task complexity. For this reason, most work focuses on training in simulation before transferring to real robots. We have reached a point where multiple days or even weeks are needed to fully train an agent with current simulators. For example, OpenAI's block reorientation task was trained for up to 14 days and their Rubik's cube solving policy took several months to train [4]. The problem is exacerbated by the fact that deep

5th Conference on Robot Learning (CoRL 2021), London, UK.

reinforcement learning requires hyper-parameter tuning to obtain a suitable solution which requires sequentially rerunning time-consuming training. Reducing training times using massively parallel approaches such as presented here can therefore help improve the quality and time-to-deployment of DRL policies, as a training setup can be iterated on more often in the same time frame.

In this paper, we examine the effects of massive parallelism for on-policy DRL algorithms and present considerations in how the standard RL formulation and the most commonly used hyper-parameters should be adapted to learn efficiently in the highly parallel regime. Additionally, we present a novel game-inspired curriculum which automatically adapts the task difficulty to the performance of the policy. The proposed curriculum architecture is straightforward to implement, does not require tuning, and is well suited for the massively parallel regime. Common robotic simulators such as Mujoco [5], Bullet [6], or Raisim [7] feature efficient multi-body dynamics implementations. However, they have been developed to run on CPUs with only a reduced amount of parallelism. In this work, we use NVIDIA's Isaac Gym simulation environment [8], which runs both the simulation and training on the GPU and is capable of simulating thousands of robots in parallel.

The massively parallel training regime has been explored before [4, 9] in the context of distributed systems with a network of thousands of CPUs each running a separate instance of the simulation. The parallelization was achieved by averaging the gradients between the different workers without reducing the number of samples provided by each agent. This results in large batch sizes of millions of samples for each policy update which improves the learning dynamics, but does not optimize the overall training time. In parallel, recent works have aimed to increase the simulation through-put and reduce training times of standard DRL benchmark tasks. A framework combining parallel simulation with multi-GPU training [10] was proposed to achieve fast training using hundreds of parallel agents. In the context of visual navigation, large batch simulation has been used to increase the training throughput [11]. Furthermore, GPU accelerated physics simulation has been shown to significantly improve the training time of the Humanoid running task [12]. A differentiable simu-lator running on Google's TPUs has also been shown to greatly accelerate the training of multiple tasks [13]. We build upon [10, 12] by pushing the parallelization further, optimizing the training algorithm, and applying the approach to a challenging real-world robotics task.

Perceptive and dynamic locomotion for legged robots in unstructured environments is a demanding task that, until recently, had only been partially demonstrated with complex model-based approaches [14, 15]. Learning-based approaches are emerging as a promising alternative. For quadrupeds, DRL has been used to train blind policies robust to highly uneven ground [16] (12 hours of training). Per-ceptive locomotion over challenging terrain has been achieved by combining learning with optimal control techniques [17, 18] (82 and 88 hours of training) and recently, a fully learned approach has shown great robustness in this setting [19] (120 hours of training). Similarly, bipedal robots have also been trained to walk blindly on stairs [20] (training time not reported). With our approach we can train a perceptive policy in under 20 minutes on a single GPU, with the complexity of sim-to-real transfer to the hardware, which increases the performance and robustness requirements and provides clear validation of the overall approach. Training such behaviors in minutes opens up new exciting possibilities ranging from automatic tuning to customized training using scans of particular environments.

## 2 Massively Parallel Reinforcement Learning

Current (on-policy) reinforcement learning algorithms are divided into two parts: data collection and policy update. The policy update, which corresponds to back-propagation for neural networks, is easily performed in parallel on the GPU. Parallelizing data collection is not as straightforward. Each step consists of policy inference, simulation, reward, and observation calculation. Current popular pipelines have the simulation and reward/observation calculation computed on the CPU, making the GPU unsuitable for policy inference because of communication bottle-necks. Data transfer over PCIe is known to be the weakest link of GPU acceleration, and can be as much as 50 times slower than the GPU processing time alone [21]. Furthermore, with CPU data collection, a large amount of data must be sent to the GPU for each policy update, slowing down the overall process. Limited parallelization can be achieved by using multiple CPU cores and spawning many processes, each running the simulation for one agent. However, the number of agents is quickly limited by the num-ber of cores and other issues such as memory usage. We explore the potential of massive parallelism

with Isaac Gym's end-to-end data collection and policy updates on the GPU, significantly reducing data copying and improving simulation throughput.

## 2.1 Simulation Throughput

The main factor affecting the total simulation throughput is the number of robots simulated in parallel. Modern GPUs can handle tens of thousands of parallel instructions. Similarly, IsaacGym's PhysX engine can process thousands of robots in a single simulation and all other computations of our pipeline are vectorized to scale favorably with the number of robots. Using a single simulation with thousands of robots presents some new challenges. For example, a single common terrain mesh must be used, and it cannot be easily changed at each reset. We circumvent this problem by creating the whole mesh with all terrain types and levels tiled side by side. We change the terrain level of the robots by physically moving them on the mesh. In supplementary material, we show the computational time of different parts of the pipeline, examine how these times scale with the number of robots, and provide other techniques to optimize the simulation throughput.

## 2.2 DRL Algorithm

We build upon a custom implementation of the Proximal Policy Optimization (PPO) algorithm [22]. Our implementation is designed to perform every operation and store all the data on the GPU. In order to efficiently learn from thousands of robots in parallel, we perform some essential modifications to the algorithm and change some of the commonly used hyper-parameter values.

### 2.2.1 Hyper-Parameters Modification

In an on-policy algorithm such as PPO, a fixed policy collects a selected amount of data before doing the next policy update. This *batch size, B*, is a crucial hyper-parameter for successful learning. With too little data, the gradients will be too noisy, and the algorithm will not learn effectively. With too much data, the samples become repetitive, and the algorithm cannot extract more information from them. These samples represent wasted simulation time and slow down the overall training. We have $B = n_{robots} n_{steps}$, where $n_{steps}$ is the number of steps each robot takes per policy update and $n_{robots}$ the number of robots simulated in parallel. Since we increase $n_{robots}$ by a few orders of magnitude, we must choose a small $n_{steps}$ to keep $B$ reasonable and hence optimize training times, which is a setting that has not been extensively explored for on-policy reinforcement learning algorithms. It turns out that we can not choose $n_{steps}$ to be arbitrarily low. The algorithm requires trajectories with coherent temporal information to learn effectively. Even though, in theory, information of single steps could be used, we find that the algorithm fails to converge to the optimal solution below a certain threshold. This can be explained by the fact that we use Generalized Advantage Estimation (GAE) [23], which requires rewards from multiple time steps to be effective. For our task, we find that the algorithm struggles when we provide fewer than 25 consecutive steps, corresponding to $0.5\,\mathrm{s}$ of simulated time. It is important to distinguish $n_{steps}$ from the maximum episode length leading to a time-out and a reset, which we define as $20\,\mathrm{s}$. The environments are reset when they reach this maximum length and not after each iteration, meaning that a single episode can cover many policy updates. This limits the total number of robots training in parallel, and consequently, prohibits us from using the full computational capabilities of the GPU.

The *mini-batch size* represents the size of the chunks in which the *batch size* is split to perform back-propagation. We find that having mini-batch sizes much larger than what is usually considered best practice is beneficial for our massively parallel use case. We use mini-batches of tens of thousands of samples and observe that it stabilizes the learning process without increasing the total training time.

### 2.2.2 Reset Handling

During training, the robots must be reset whenever they fall, and also after some time to keep them exploring new trajectories and terrains. The PPO algorithm includes a *critic* predicting an infinite horizon sum of future discounted rewards. Resets break this infinite horizon assumption and can lead to inferior critic performance if not handled carefully. Resets based on failure or reaching a goal are not a problem because the critic can predict them. However, a reset based on a time out can not be predicted (we do not provide episode time in the observations). The solution is to distinguish

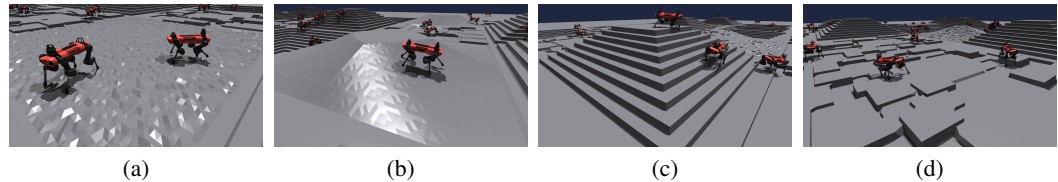

|        |        |        |        |
|:------:|:------:|:------:|:------:|
| (a)    | (b)    | (c)    | (d)    |

Figure 2: Terrain types used for training and testing in simulation. (a) Randomly rough terrain with variations of $0.1\,\mathrm{m}$. (b) Sloped terrain with an inclination of $25\,\mathrm{deg}$. (c) Stairs with a width of $0.3\,\mathrm{m}$ and height of $0.2\,\mathrm{m}$. (d) Randomized, discrete obstacles with heights of up to $\pm 0.2\,\mathrm{m}$.

the two termination modes and augment the reward with the expected infinite sum of discounted future rewards in a time-out case. In other words, we bootstrap the target of the critic with its own prediction. This solution has been discussed in [24], but interestingly, this distinction is not part of the widely used *Gym* environment interface [25] and is ignored by popular implementations such as *Stable-Baselines* [26][1]. After investigating multiple implementations, we conclude that this important detail is often avoided by assuming that the environments either never time out or only on the very last step of a batch collection. In our case, with few robot steps per batch, we can not make such an assumption since a meaningful episode length covers the collection of many batches. We modify the standard *Gym* interface to detect time-outs and implement the bootstrapping solution. In supplementary material, we show the effect of this solution on the total reward as well as the critic loss.

## 3    Task Description

A quadruped robot must learn to walk across challenging terrain, including uneven surfaces, slopes, stairs, and obstacles, while following base-heading and linear-velocity commands. We conduct most of the simulation and real-world deployment experiments on the ANYbotics ANYmal C robot. However, in simulation, we demonstrate the broader applicability of the approach by additionally training policies for ANYmal B, ANYmal C with an attached arm, and the Unitree A1 robots.

### 3.1    Game-Inspired Curriculum

The terrains are selected to be representative of real-world environments. We create five types of procedurally generated terrains presented in Fig. 2: flat, sloped, randomly rough, discrete obstacles, and stairs. The terrains are tiled squares with 8m sides. The robots start at the center of the terrain and are given randomized heading and velocity commands (kept constant for the duration of an episode) pushing them to walk across the terrain. Slopes and stairs are organized in pyramids to allow traversability in all directions.

Previous works have shown the benefits of using an automated curriculum of task difficulty to learn complex locomotion policies [28, 29, 16]. Similarly, we find that it is essential to first train the policy on less challenging terrain before progressively increasing the complexity. We adopt a solution inspired by [16], but replace the particle filter approach with a new game-inspired automatic curriculum. All robots are assigned a terrain type and a level that represents the difficulty of that terrain. For stairs and randomized obstacles, we gradually increase the step height from $5\,\mathrm{cm}$ to $20\,\mathrm{cm}$. Sloped terrain inclination is increased from $0\,\mathrm{deg}$ to $25\,\mathrm{deg}$. If a robot manages to walk past the borders of its terrain, its level is increased, and at the next reset, it will start on more difficult terrain. However, if at the end of an episode it moved by less than half of the distance required by its target velocity, its level is reduced again. Robots solving the highest level are looped back to a randomly selected level to increase the diversity and avoid catastrophic forgetting. This approach has the advantage of training the robots at a level of difficulty tailored to their performance without requiring any external tuning. It adapts the difficulty level for each terrain type individually and provides us with visual and quantitative feedback on the progress of the training. When the robots have reached the final level and are evenly spread across all terrains due to looping back, we can conclude they have fully learned to solve the task.

---

[1] The *Spinning-up* [27] implementation of PPO uses the same bootstrapping solution by keeping track of episode lengths within the algorithm, thus circumventing the limitation of the *Gym* interface.

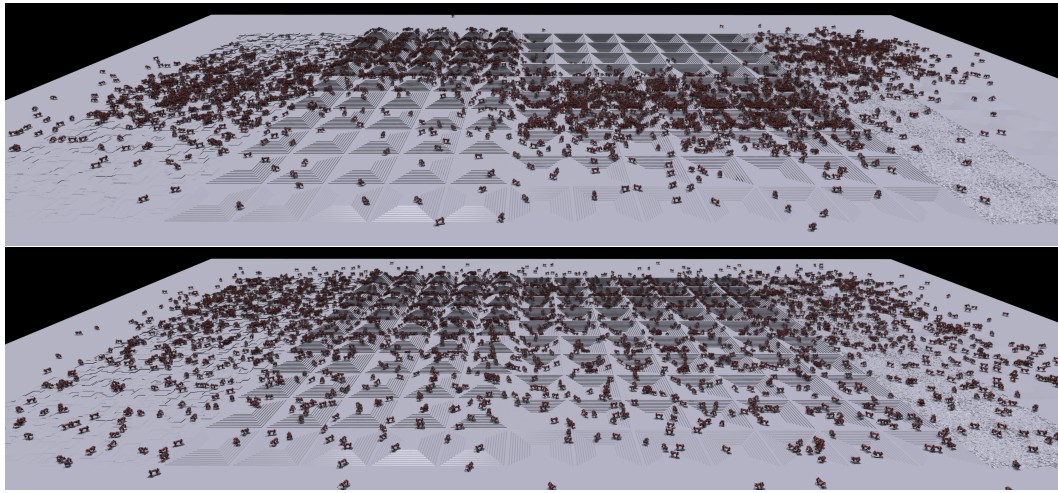

Figure 3: 4000 robots progressing through the terrains with automatic curriculum, after 500 (top) and 1000 (bottom) policy updates. The robots start the training session on the first row (closest to the camera) and progressively reach harder terrains.

The proposed curriculum structure is well suited for the massively parallel regime. With thousands of robots we can directly use their current progress in the curriculum as the distribution of the policy's performance, and do not need learn it with a generator network [30]. Furthermore, our method doesn't require tuning and is straightforward to implement in a parallel manner with near-zero processing cost. We remove the computational overhead of re-sampling and re-generating new terrains needed for the particle filter approach.

Fig. 3 shows robots progressing through the terrains at two different stages of the training process. On complex terrain types, the robots require more training iterations to reach the highest levels. The distribution of robots after 500 iterations shows that while the policy is able to cross sloped terrains and to go down stairs, climbing stairs and traversing obstacles requires more training iterations. However, after 1000 iterations, the robots have reached the most challenging level for all terrain types and are spread across the map. We train for a total for 1500 iterations to let the policy converge to its highest performance.

## 3.2    Observations, Actions, and Rewards

The policy receives proprioceptive measurements of the robot as well as terrain information around the robot's base. The observations are composed of: base linear and angular velocities, measurement of the gravity vector, joint positions and velocities, the previous actions selected by the policy, and finally, 108 measurements of the terrain sampled from a grid around the robot's base. Each measurement is the distance from the terrain surface to the robot's base height.

The total reward is a weighted sum of nine terms, detailed in supplementary material. The main terms encourage the robot to follow the commanded velocities while avoiding undesired base velocities along other axes. In order to create a smoother, more natural motion, we also penalize joint torques, joint accelerations, joint target changes, and collisions. Contacts with the knees, shanks or between the feet and a vertical surface are considered collisions, while contacts with the base are considered crashes and lead to resets. Finally, we add an additional reward term encouraging the robot to take longer steps, which results in a more visually appealing behavior. We train a single policy with the same rewards for all terrains.

The actions are interpreted as desired joint positions sent to the motors. There, a PD controller produces motor torques. In contrast to other works [16, 20], neither the reward function nor the action space has any gait-dependent elements.

## 3.3    Sim-to-Real Additions

In order to make the trained policies amenable for sim-to-real transfer, we randomize the friction of the ground, add noise to the observations and randomly push the robots during the episode to

teach them a more stable stance. Each robot has a friction coefficient sampled uniformly in [0.5, 1.25]. The pushes happen every $10\,\mathrm{s}$. The robots' base is accelerated up to $\pm 1\,\mathrm{m/s}$ in both x and y directions. The amount of noise is based on real data measured on the robot and is detailed in supplementary material.

The ANYmal robot uses series elastic actuators with fairly complex dynamics, which are hard to model in simulation. For this reason and following the methodology of previous work [1], we use a neural network to compute torques from joint position commands. However, we simplify the inputs of the model. Instead of concatenating past measurements at fixed time steps and sending all of that information to a standard feed-forward network, we only provide the current measurements to an LSTM network. A potential drawback of this set-up is that the policy does not have the temporal information of the actuators as in previous work. We have experimented with various ways of providing that information through memory mechanisms for the policy but found that it does not improve the final performance.

## 4 Results

### 4.1 Effects of Massive Parallelism

In this section, we study the effects of the number of parallel robots on the final performance of the policy. In order to use the total reward as a single representative metric, we have to remove the curriculum, otherwise a more performant policy sees its task difficulty increase and consequently a decrease in the total reward. As such, we simplify the task by reducing the maximum step size of stairs and obstacles and directly train robots on the full range of difficulties.

We begin by setting a baseline with $n_{robots} = 20000$ and $n_{steps} = 50$, resulting in a batch size of 1M samples. Using this very large batch size results in the best policy but at the cost of a relatively long training time.

We then conduct experiments in which we increase the number of robots while keeping the batch size constant. As a result, the number of steps each robot takes per policy update decreases. In this case, the training time decreases with a higher number of robots, but the policy performance drops if that number is too high. We start from 128 robots corresponding to the level of parallelization of previous CPU implementations and increase that number up to 16384, which is close to the maximum amount of robots we could simulate on rough terrain with Isaac Gym running on a single workstation GPU.

In Fig. 4, we compare these results with the baseline, which allows us to select the most favorable trade-off between policy performance and training time. We see two interesting effects at play. First, when the number of robots is too high, the performance drops sharply, which can be explained by the time horizon of each robot becoming too small. As expected, with larger batch sizes, the overall reward is higher, and the time horizon effect is shifted, meaning that we can use more robots before seeing the drop. On the other hand, below a certain threshold, we see a slow decrease in

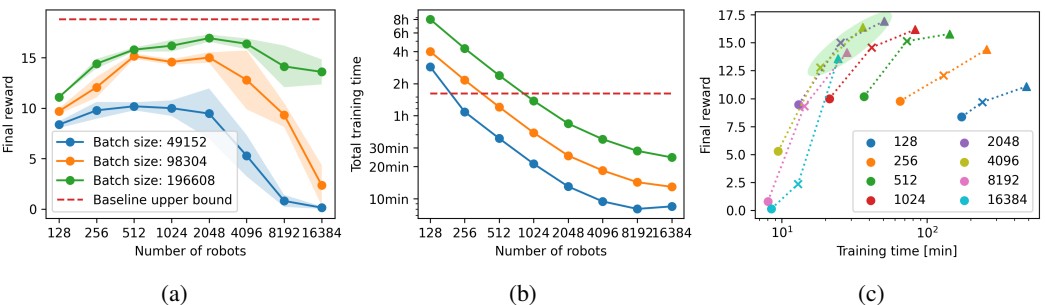

Figure 4: (a) Average and standard deviation (over 5 runs) of the total reward of an episode after 1500 policy updates for different number of robots and 3 different batch sizes. The ideal case of a batch size of 1M samples with 20000 robots is shown in red. (b) Total training time for the same experiments. (c) Reward dependency on total training time. Colors represent the number of robots, while shapes show the batch size (circles: 49152, crosses: 98304, triangles: 196608). Points in the upper left part of the graph (highlighted in green) represent the most desirable configuration.

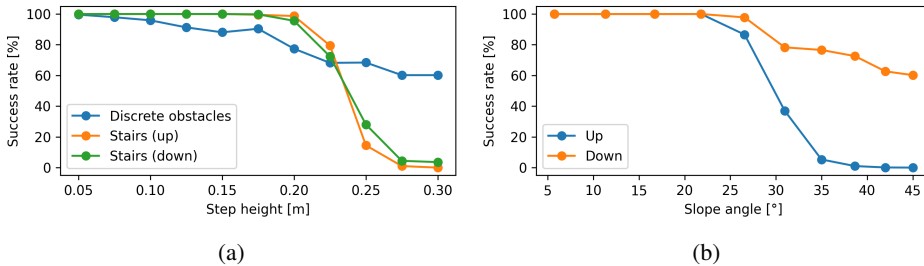

|     | (a) | | (b) |

Figure 5: Success rate of the tested policy on increasing terrain complexities. Robots start in the center of the terrain and are given a forward velocity command of $0.75\,\mathrm{m/s}$, and a side velocity command randomized within $[-0.1, 0.1]\,\mathrm{m/s}$. (a) Success rate for climbing stairs, descending stairs and traversing discrete obstacles. (b) Success rate for climbing and descending sloped terrains.

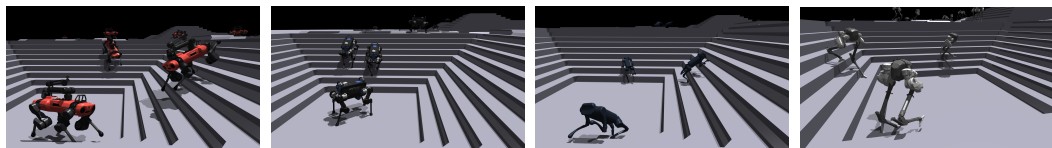

Figure 6: ANYmal C with a fixed arm, ANYmal B, A1 and Cassie in simulation.

performance with fewer robots. We believe this is explained by the fact that the samples are very similar with many steps per robot because of the relatively small time steps between them. This means that for the same amount of samples, there is less diversity in the data. In other words, with a low number of robots, we are further from the standard assumption that the samples are independent and identically distributed, which seems to have a noticeable effect on the training process. In terms of training time, we see a nearly linear scaling up to 4000 robots, after which simulation throughput gains slow down. As such, we can conclude that increasing the number of robots is beneficial for both final performance and training time, but there is an upper limit on this number after which an on-policy algorithm cannot learn effectively. Increasing the batch size to values much larger than what is typically used in similar works seems highly beneficial. Unfortunately, it also scales the training time so it is a trade-off that must be balanced. From the third plot we can conclude that using 2048 to 4096 robots with a batch size of $\approx 100k$ or $\approx 200k$ provides the best trade-off for this specific task.

## 4.2 Simulation

For our simulation and deployment experiments, we use a policy trained with 4096 robots and a batch size of 98304, which we train for 1500 policy updates in under 20 minutes[2]. We begin by measuring the performance of our trained policy in simulation. To that end, we perform robustness and traversability tests. For each terrain type, we command the robots to traverse the representative difficulty of the terrain at high forward velocity and measure the success rate. A success is defined as managing to cross the terrain while avoiding any contacts on the robot's base. Fig. 5 shows the results for the different terrains. For stairs, we see a nearly $100\,\%$ success rate for steps up to $0.2\,\mathrm{m}$, which is the hardest stair difficulty we train on and close to the kinematic limits of our robot. Randomized obstacles seem to be more demanding, with the success rate decreasing steadily. We must note that in this case, the largest step is double the reported height since neighboring obstacles can have positive and negative heights. In the case of slopes, we can observe that after $25\,\mathrm{deg}$ the robots are not able to climb anymore but still learn to slide down with a moderate success rate.

Given our relatively simple rewards and action space, the policy is free to adopt any gait and behavior. Interestingly, it always converges to a trotting gait, but there are often artifacts in the behavior, such as a dragging leg or unreasonably high or low base heights. After tuning of the reward weights, we can obtain a policy that respects all our constraints and can be transferred to the physical robot.

To verify the generalizability of the approach, we train policies for multiple robots with the same set-up. We use the ANYmal C robot with a fixed robotic arm, which adds about $20\,\%$ of additional

---

[2]Trained on: i9-11900k CPU, NVIDIA RTX A6000 GPU. VRAM requirements are in the supplementary material.

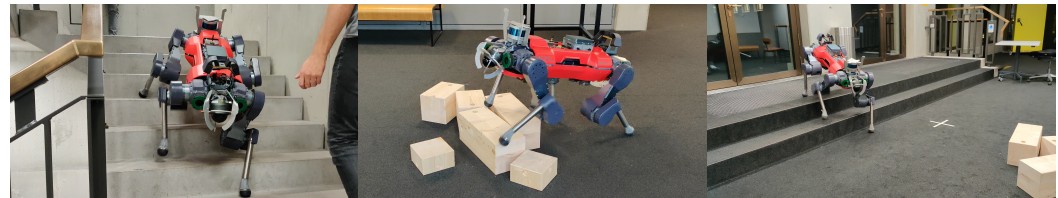

Figure 7: Locomotion policy, trained in under 20min, deployed on the physical robot.

weight, and the ANYmal B robot, which has comparable dimensions but modified kinematic and dynamic properties. In these two cases, we can retrain a policy without any modifications to the rewards or algorithm hyper-parameters and obtain a very similar performance. Next, we use the Unitree A1 robot, which has smaller dimensions, four times lower weight, and a different leg configuration. In this case, we remove the actuator model of the ANYdrive motors, reduce PD gains and the torque penalties, and change the default joint configurations. We can train a dynamic policy that learns to solve the same terrains even with the reduced size of the robot. Finally, we apply our approach to Agility Robotics' bipedal robot Cassie. We find that an additional reward encouraging standing on a single foot is necessary to achieve a walking gait. With this addition, we are able to train the robot on the same terrains as its quadrupedal counterparts. Fig. 6 shows the different robots.

### 4.3 Sim-to-real Transfer

On the physical robot, our policy is fixed. We compute the observations from the robot's sensors, feed them to the policy, and directly send the produced actions as target joint positions to the motors. We do not apply any additional filtering or constraint satisfaction checks. The terrain height measurements are queried from an elevation map that the robot is building from Lidar scans.

Unfortunately, this height map is far from perfect, which results in a decrease in robustness between simulation and reality. We observe that these issues mainly occur at high velocities and therefore reduce the maximum linear velocity commands to $0.6\,\mathrm{m/s}$ for policies deployed on the hardware. The robot can walk up and down stairs and handles obstacles in a dynamic manner. We show samples of these experiments in Fig. 7 and in the supplementary video. To overcome issues with imperfect terrain mapping or state estimation drift, the authors of [19] implemented a teacher-student set-up, which provided outstanding robustness even in adverse conditions. As part of future work, we plan to merge the two approaches.

## 5  Conclusion

In this work, we demonstrated that a complex real-world robotics task can be trained in minutes with an on-policy deep reinforcement learning algorithm. Using an end-to-end GPU pipeline with thousands of robots simulated in parallel, combined with our proposed curriculum structure, we showed that the training time can be reduced by multiple orders of magnitude compared to previous work. We discussed multiple modifications to the learning algorithm and the standard hyper-parameters required to use the massively parallel regime effectively. Using our fast training pipeline, we performed many training runs, simplified the set-up, and kept only essential components. We showed that the task can be solved using simple observation and action spaces as well as relatively straightforward rewards without encouraging particular gaits or providing motion primitives.

The purpose of this work is not to obtain the absolute best-performing policy with the highest robustness. For that use case, many other techniques can be incorporated into the pipeline. We aim to show that a policy can be trained in record time with our set-up while still being usable on the real hardware. We wish to shift other researchers' perspective on the required training time for a real-world application, and hope that our work can serve as a reference for future research. We expect many other tasks to benefit from the massively parallel regime. By reducing the training time of these future robotic tasks, we can greatly accelerate the developments in this field.

**Acknowledgments**

We would like to thank Mayank Mittal, Joonho Lee, Takahiro Miki, and Peter Werner for their valuable suggestions and help with hardware experiments as well as the Isaac Gym and PhysX teams for their continuous support.

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
