# OpenReview forum: "Learning to Walk in Minutes Using Massively Parallel Deep Reinforcement Learning"
_robot-learning.org/CoRL/2021/Conference — CoRL2021 Poster_

### Official Review · Reviewer_Y3ip · 2021-07-17

**Originality:** Good
**Technical Quality:** Very Good
**Clarity Of Presentation:** Very Good
**Impact:** 3

**Recommendation:**

Weak Accept: I recommend accepting the paper, but will not argue for my recommendation if the majority of other reviewers have a different opinion.

**Summary:**

Main Contributions
================
A new training setup for model-free reinforcement learning that allows training policies for quadruped robots very efficiently.
An ablation study that investigates the impact of batch size and number of robots simulated in parallel on final performance.

Summary
========
The paper demonstrates a learning strategy for quadruped locomotion that is efficiently implemented entirely on the GPU and allows massive parallelization during training time.
Combined with a game-inspired curriculum learning, the policies successfully learn to navigate complex terrain and also transfer to a real robot.
In the experimental part, the paper presents an ablation study of batch size and number of robots on the final performance and training time required.




**Issues:**

The paper would be stronger if the game-inspired curriculum would be compared to the previous, particle-filter based approach of curriculum learning.

**Reviewer Expertise:**

Good: General knowledge of the area

**Strengths And Weaknesses:**

Strengths
=========
- Nice video!
- The paper is well written.
- This training methodology would be very useful if made open-source, as it might be generalized to other robotic platforms as well.

Weaknesses
===========
- The paper contains very limited scientific novelty. Previous works [1, 2] have already demonstrated the feasibility of training a navigation policy for a quadruped robot in simulation and deployment in the real world.
- The game-inspired curriculum is the main novelty of this work, but is not compared to the previous approach used for training (particle filter approach proposed in [1]).


[1] Lee, J., Hwangbo, J., Wellhausen, L., Koltun, V. and Hutter, M., 2020. Learning quadrupedal locomotion over challenging terrain. Science robotics, 5(47).

[2] Hwangbo, J., Lee, J., Dosovitskiy, A., Bellicoso, D., Tsounis, V., Koltun, V. and Hutter, M., 2019. Learning agile and dynamic motor skills for legged robots. Science Robotics, 4(26).

**Summary Of Recommendation:**

The usability of this work is highly dependent on the fact if the code will be available open-source. Due to the very limited scientific novelty (i.e. the game-inspired curriculum), the main purpose of this paper would be to publicize a new highly parallelizable training framework.

---

> ### Author Response · Authors · 2021-08-26
> **Response to Reviewer Y3ip**
>
> We thank the reviewer for their interest in our code and for raising important concerns. In terms of the task, we build upon [1] by adding perceptive inputs which allow the policy to better overcome obstacles. A blind policy detects the presence of an obstacle by hitting it, while with perception it can be overcome without impact.
> The main novelty of our approach is the vastly faster training times achieved with the new training procedure. We show that these tasks can be trained in minutes instead of hours or even days. We optimize the training algorithm and provide insights into the effect of massive parallelism on the policy's performance.
> We hope that by providing the full code of the environment as well as the algorithm, our work can have an important impact by helping other researchers reduce the iteration time of their training tasks and as such speed up the developments in this field.
>
> We understand the desire for a comparison between the two curriculum approaches and have explored ways of providing it. Unfortunately implementing the same particle filter as in [1] would be impractical in our pipeline. It would require the regeneration of terrains at each step which would dramatically decrease the simulation throughput (adding a new mesh to the GPU simulation can take seconds). We have explored the idea of implementing a simplified one-dimensional particle filter that would select between our discrete terrains, but we have concluded that this would bring us back very close to our current implementation, and as such any comparison would be of limited use. However, our curriculum is simpler to implement, does not require any additional tuning, and can be applied to the massively parallel regime with nearly zero additional computational cost. We have added a few sentences detailing this in the paper.

---

> > ### Comment · Reviewer_Y3ip · 2021-09-02
> > **Post-rebuttal reponse**
> >
> > I thank the authors for the clarification and maintain my score of 'weak accept' due to the limited scientific novelty.

---

### Official Review · Reviewer_d4o6 · 2021-07-23

**Originality:** Good
**Technical Quality:** Very Good
**Clarity Of Presentation:** Very Good
**Impact:** 4

**Recommendation:**

Strong Accept: I recommend accepting the paper and will argue for my recommendation even if other reviewers hold a different opinion.

**Summary:**

The paper explores learning locomotion policies for quadruped robots using massive parallel simulation. The main technical contributions of the paper is an adapted setting of the PPO algorithm for massive parallel simulations, as well as a curriculum learning scheme for training the policies to tackle complex environments. They also demonstrated that the resulting control policy can be successfully deployed to the real robot.


**Issues:**

As mentioned in the weakness part, I think there are three issues I’d like to see more discussions/experiments:

1. How is the perturbation applied to the robot during training? Is the robot reset if the episode ends due to time-out?
2. Add discussion on how the work differs from [10] and ideally experiments to show that the delta in the learning algorithm makes a significant difference.
3. It would be great to have more diverse behaviors emerge from this nice framework, for example biped locomotion or manipulation.


**Reviewer Expertise:**

Excellent: Expert knowledge on the topic of the paper

**Strengths And Weaknesses:**

Strengths:
The main strength of the paper is that it demonstrates the possibility of leveraging large-scale simulations to learn legged locomotion for a real robot in a very short amount of time. The modifications to the standard PPO setup in response to the issues arising from the massive parallelization is reasonable and supported by their experiments. The proposed curriculum scheme also makes sense.


Weaknesses:
1. Some details in the paper are not clear. For example, the paper mentioned that the episode length for each simulation is short (<5s) due to the large number of robots being simulated. At the end of the episode, is the robot reset to the initial state? The paper also mentioned that a random push force is applied every 10s to the robot. How is this 10s computed since it will span multiple episodes?
2. In [10], the GPU-based simulation was also used to learn several robotic tasks in a short amount of time, similar to the presented work. It’s not very clear what’s the main difference between the two works in terms of large-scale training. For example, if one takes the learning algorithm from [10], applies the curriculum learning scheme and reward function in this work, what would the result be like?
3. The proposed method converged to the same locomotion gait (trotting) for all three robots tested. This is good for real robot transfer given the stability of trotting gaits, meanwhile the lack of diverse behaviors in the robots is a bit unsatisfying. It would be nice if more locomotion forms can be tested such as bipedal walking.


**Summary Of Recommendation:**

The paper presents an interesting direction in learning robot locomotion controllers with massively parallelized simulations. The result is solid in general with real robot validations and can inspire further research in this direction. The several technical details introduced in the paper to help improve training performance is also reasonable and effective. However, there are a few issues with the paper such as missing details, limited discussion with relevant work, and limited diversity in the learned behaviors. It would be a good contribution to CoRL if those can be addressed.

---

> ### Author Response · Authors · 2021-08-26
> **Response to Reviewer d4o6**
>
> Thank you for pointing out imprecisions in the paper and for the interesting suggestions.
> 1. We distinguish episode length from the length of a trajectory used for each iteration of PPO. In our experiments, we collect trajectories of ~0.5s before updating the policy. However, the environments are not reset after an update and the episodes continue. The episode length leading to a time-out is 20 seconds in our case. We have added this explanation to the paper.
>
> 2. Our work builds upon [10] (now [12] in the paper) by pushing the parallelization even further and implementing important additions to the PPO algorithm. We believe that the algorithm of [10] can be used to solve our task, but it would lead to slower convergence without our optimizations. Furthermore, in this paper, we aim to show the applicability of this parallel approach to a real-world task, by using an existing robot, including our sim-to-real additions and transferring trained policies to the real world.
>
> 3. We have applied our approach to the bipedal robot Cassie and have obtained great results which demonstrates the benefits of the presented approach. The experience of training a completely new robot without any prior knowledge about its specific structure has been very interesting. After half a day of work to import the model, adapt the urdf, and tune default joint positions and gains, it took us a day and around 40 training runs to obtain a good walking gait on flat terrain (we ended up needing an additional reward term encouraging single foot contacts to avoid pronking behaviors). Once we had good results on flat terrain, we could easily train a policy for perceptive/rough terrain locomotion without any additional changes. Overall we were able to train our final policy within 2 days of work. We have added these results to the paper and the video.

---

> > ### Comment · Reviewer_d4o6 · 2021-09-01
> > **Post-rebuttal reponse**
> >
> > The response and additional experiments from the authors are greatly appreciated. The rebuttal addressed my questions regarding the training detail and the cassie experiment is quite interesting. Thus I have increased my score.
> >
> > On the other hand, I still believe it would be valuable to explicitly compare to [12]. Without proper comparison, the paper currently is more like applying the idea to real-world robotics tasks, which diminishes the potential impact of the proposed algorithm.

---

### Official Review · Reviewer_cSHN · 2021-07-24

**Originality:** Fair
**Technical Quality:** Good
**Clarity Of Presentation:** Very Good
**Impact:** 3

**Recommendation:**

Weak Accept: I recommend accepting the paper, but will not argue for my recommendation if the majority of other reviewers have a different opinion.

**Summary:**

The paper demonstrates that it is possible to use a GPU-based physics simulator to run thousands of simulations in parallel, which can greatly accelerate the DRL training of quadruped locomotion. Batch size hyper-parameters are tuned to adapt to this setting, and a curriculum of "different levels" with gradually increasing terrain difficulty is designed. The learned policy can also be deployed on a real robot, showing its robustness.

**Issues:**

As discussed, code release will make this paper more consistent with the spirit of CoRL.

Question:
Is it possible to reset simulation to previously visited states, rather than always close to the starting pose? Seems like if so we can worry less about number of consecutive steps being too small?

**Reviewer Expertise:**

Very good: Comprehensive knowledge of the area

**Strengths And Weaknesses:**

The paper's main strength is the quality of results -- as the authors claimed, it was not previously possible to train quadruped locomotion on a single workstation within minutes. The presentation of the material is also pretty clear.

There are two apparent weaknesses for papers like this. 1. lack of technical novelty and 2. lack of open-source implementation. The reviewer would especially like to emphasize the importance of 2 -- CoRL has long been advocating for open-sourcing; if the authors do want to publish their findings at CoRL, they should seriously consider packaging and releasing their implementation to help real advance of the community.

Related work (GPU for massively parallel DRL, with open-source code: https://graphics.stanford.edu/projects/bps3D/)

**Summary Of Recommendation:**

The results of the paper look impressive. The main reason for "weak" accept is that without open-sourced code, it is difficult to validate how fast the implementation really is.

---

> ### Author Response · Authors · 2021-08-26
> **Response to Reviewer cSHN**
>
> We thank the reviewer for their interest in our code and have decided to release both the environments as well as the training algorithms. Please find our code in the supplementary material. If the paper is accepted, we will make the code publicly available. We hope that other researchers will be able to accelerate their respective training pipelines, contribute to the development of new tasks and algorithms, and ultimately extend the capabilities of legged robots.
> It is possible to reset the simulation to any arbitrary state. However, we must note that we do not reset the simulation after each learning iteration, which means the robots continue their respective trajectories even though the policy is updated. Unfortunately, this is not sufficient to fully overcome the problem of short time horizons. We believe this is due to the Generalizer Advantage Estimate (GAE) requiring multiple consecutive time steps to be effective. We have made this explanation more precise in the paper.
> We have added a reference to the relevant suggested work.

---

> > ### Comment · Reviewer_cSHN · 2021-08-31
> > **Reply**
> >
> > I am happy to see the open-source code coming out. Maintaining my "weak accept" rating on basis of marginal novelty.

---

### Official Review · Reviewer_2XwF · 2021-07-28

**Originality:** Good
**Technical Quality:** Good
**Clarity Of Presentation:** Very Good
**Impact:** 3

**Recommendation:**

Weak Accept: I recommend accepting the paper, but will not argue for my recommendation if the majority of other reviewers have a different opinion.

**Summary:**

This paper presents an experimental setup for learning locomotion controllers for  legged robots, with an emphasis on robotic quadrupeds. The system consists of a simulation pipeline on which multiple robot simulations can be run in parallel, along with a policy optimization algorithm that combines data from all simulations. The proposed simulation environment is shared by multiple instances of simulated robots. The environment is divided into tiles with different terrains, which pose challenges of varying difficulties to a walking robot. This setup is used to design a curriculum where robots are assigned a tile, a desired heading and desired velocity, and are promoted/demoted to harder/easier environments depending on whether they exit the tile at the desired heading. The paper shows how this setup can be used to train walking controllers with online policy search (PPO) on a single GPU (with 48gb of VRAM)

**Issues:**

Some references to related work on curriculum learning for locomotion [1] and [2], and on adapting PPO to parallel simulation [3].

[1] Paired Open-Ended Trailblazer (POET):
[2] ALLSTEPS: Curriculum-driven Learning of Stepping Stone Skills
[3] Accelerated Methods for Deep Reinforcement Learning

**Reviewer Expertise:**

Very good: Comprehensive knowledge of the area

**Strengths And Weaknesses:**

The presentation and the engineering work done to produce the results in this paper are very itneresting. The proposed approach of large scale data parallelism on a single GPU is definitely relevant for robotics research.

It is not clear whether the technique for reset handling described in section 2.2.2 is doing anything different from existing implementations of PPO. Reset handling as described in this section is already present in other implementations of PPO (e.g. in openai baselines and the spinningup repositories), which use the standard gym interface.

Since the paper makes an emphasis on training on a single GPU, it would be very useful to have an evaluation in terms of available GPU RAM given that not many practitioners have access to the GPU used in this work.



**Summary Of Recommendation:**

This is a neat demonstration of how existing algorithms for policy search can benefit from scaling up data collection in simulation. This paper describes the engineering decision taken to enable training on a single GPU.

---

> ### Author Response · Authors · 2021-08-26
> **Response to Reviewer 2XwF**
>
> We thank the reviewer for the valuable comments and suggestions.
> An important distinction of our implementation is the fact that time-outs can happen at any step and individually for each robot. In the popular baselines and stable-baselines implementations, there is no distinction between a reset due to a termination condition or a time out. However, if the timeouts all happen at the first step of each iteration, (or don’t happen at all) this distinction is not needed. Interestingly, the spinning-up implementation does make the distinction and explicitly checks for timeouts at each step. Our Implementation is similar to spinning-up, but we let the environment decide when to timeout and implement the bootstrapping in a parallelized way. We have added a comment to the paper mentioning the spinning-up implementation.
> GPU RAM is indeed an important consideration. We have added a plot of RAM usage for different numbers of environments to the supplementary material. We can see that 9Gb is sufficient to train 4000 robots on rough terrain with graphics and 6Gb without rendering. As such it is possible to run the training on consumer-grade GPUs such as 3080 or 2080 Ti. If rendering is only enabled at test time, a lower-end/older GPU can also be used.
>
> Not mentioning the suggested works was an oversight from our side. We have added [3] to the discussion on parallelized RL and [1, 2] to the section presenting the curriculum.

---

### Meta-Review · Area_Chair_pj1J · 2021-08-14

**Recommendation:** Accept (Poster)
**Confidence:** 4

**Metareview:**

This paper addresses the acceleration issue on the DRL training for quadruped locomotion, and presents a training set-up by using massive parallelism on a single GPU workstation, and a novel game-inspired curriculum well suited for parallel training.

I agree with reviewers that this paper is well organized and the experimental results is quite promising, while the scientific novelty seems to be limited, reviewers also concerns on the lack of open-source details.

Reviewers also pointed out several other concerns or deficiencies about the paper. I agree with these comments which might be valuable in improving the quality of the paper, mainly including:

1. Comparison or discussion to some related works, both on quadruped robot locomotion and the game-inspired curriculum learning.
2. More expeirmental details to make clear some points reviewers raised, and more locomotion forms like bipedal walking tested if possible to make the paper more solid.
3. Other minor points as reviewers mentioned.

More added:
I would like to thank authors for your carefully and detailed responses to comments raised by reviewers.
Although there are still concerns on the limited scientific novelty, the scores of this paper was improved after rebuttle.
I thus recommend this paper to be accepted.

---

> ### Author Response · Authors · 2021-08-26
> **Summary of Changes and Response to the Meta Review**
>
> We would like to thank all reviewers for their fair and constructive reviews. Based on the provided comments we will open-source our training pipeline. We provide our code in attachment* and in case of publication, we will provide a public repository.
> Reviewers have pointed us towards additional related works. We have incorporated corresponding discussions into the paper.
> Following the suggestion, we have trained the bipedal robot Cassie in our pipeline. Further details of this experiment are provided in the corresponding review answer. We believe that this is a nice demonstration of the contribution of the paper to the community.
> Finally, we have addressed the unclarities raised by the reviewers.
>
> \* Please check the README for more details about the current code version and changes that will happen before the conference.

---

### Decision · Program_Chairs · 2021-09-13

**Decision:**

Accept (Poster)

**Comment:**

This paper addresses the acceleration issue on the DRL training for quadruped locomotion, and presents a training set-up by using massive parallelism on a single GPU workstation, and a novel game-inspired curriculum well suited for parallel training.

I agree with reviewers that this paper is well organized and the experimental results is quite promising, while the scientific novelty seems to be limited, reviewers also concerns on the lack of open-source details.

Reviewers also pointed out several other concerns or deficiencies about the paper. I agree with these comments which might be valuable in improving the quality of the paper, mainly including:

1. Comparison or discussion to some related works, both on quadruped robot locomotion and the game-inspired curriculum learning.
2. More expeirmental details to make clear some points reviewers raised, and more locomotion forms like bipedal walking tested if possible to make the paper more solid.
3. Other minor points as reviewers mentioned.

More added:
I would like to thank authors for your carefully and detailed responses to comments raised by reviewers.
Although there are still concerns on the limited scientific novelty, the scores of this paper was improved after rebuttle.
I thus recommend this paper to be accepted.